# Dynamics of Inflammatory and Neurodegenerative Biomarkers after Autologous Hematopoietic Stem Cell Transplantation in Multiple Sclerosis

**DOI:** 10.3390/ijms231810946

**Published:** 2022-09-19

**Authors:** Josefine Ruder, Gianna Dinner, Aleksandra Maceski, Ernesto Berenjeno-Correa, Antonia Maria Müller, Ilijas Jelcic, Jens Kuhle, Roland Martin

**Affiliations:** 1Neuroimmunology and Multiple Sclerosis Research, Neurology Clinic, University Hospital Zurich, University Zurich, Frauenklinikstrasse 26, 8091 Zurich, Switzerland; 2Department of Clinical Research, Neurology Clinic, University Hospital Basel, Schanzenstrasse 55, 4031 Basel, Switzerland; 3Department of Medical Oncology and Hematology, University Hospital Zurich, University Zurich, Rämistrasse 100, 8091 Zurich, Switzerland

**Keywords:** autologous hematopoietic stem cell transplantation, multiple sclerosis, biomarkers

## Abstract

Autologous hematopoietic stem cell transplantation (aHSCT) is a highly efficient treatment of multiple sclerosis (MS), and hence it likely normalizes pathological and/or enhances beneficial processes in MS. The disease pathomechanisms include neuroinflammation, glial cell activation and neuronal damage. We studied biomarkers that in part reflect these, like markers for neuroinflammation (C-X-C motif chemokine ligand (CXCL) 9, CXCL10, CXCL13, and chitinase 3-like 1 (CHI3L1)), glial perturbations (glial fibrillary acidic protein (GFAP) and in part CHI3L1), and neurodegeneration (neurofilament light chain (NfL)) by enzyme-linked immunosorbent assays (ELISA) and single-molecule array assay (SIMOA) in the serum and cerebrospinal fluid (CSF) of 32 MS patients that underwent aHSCT. We sampled before and at 1, 3, 6, 12, 24 and 36 months after aHSCT for serum, as well as before and 24 months after aHSCT for CSF. We found a strong increase of serum CXCL10, NfL and GFAP one month after the transplantation, which normalized one and two years post-aHSCT. CXCL10 was particularly increased in patients that experienced reactivation of cytomegalovirus (CMV) infection, but not those with Epstein-Barr virus (EBV) reactivation. Furthermore, patients with CMV reactivation showed increased Th1 phenotype in effector memory CD4+ T cells. Changes of the other serum markers were more subtle with a trend for an increase in serum CXCL9 early post-aHSCT. In CSF, GFAP levels were increased 24 months after aHSCT, which may indicate sustained astroglia activation 24 months post-aHSCT. Other CSF markers remained largely stable. We conclude that MS-related biomarkers indicate neurotoxicity early after aHSCT that normalizes after one year while astrocyte activation appears increased beyond that, and increased serum CXCL10 likely does not reflect inflammation within the central nervous system (CNS) but rather occurs in the context of CMV reactivation or other infections post-aHSCT.

## 1. Introduction

Multiple Sclerosis (MS) is a chronic autoimmune disease of the central nervous system (CNS) with multifocal destruction of the myelin sheaths as well as axonal and neuronal damage [1]. Clinically there are two major courses of MS: most frequent is relapsing-remitting MS (RRMS), when repeated episodes of neurological dysfunction recover. Untreated RRMS often develops into a secondary progressive disease (SPMS) characterized by neurodegeneration and gradually accumulating disability. Primary progressive MS (PPMS) with a continuous disease progression from onset is seen only in a minority of patients [1]. Regarding the aetiology of MS several risk factors have been discovered including a complex genetic trait with multiple risk-conferring alleles, particularly the HLA-DR15 haplotype [2], and furthermore environmental factors like viral infections, e.g., with Epstein-Barr virus (EBV), smoking, or low vitamin D [3,4], obesity in late childhood and recently also imbalances of the gut microbiota [5].

The pathophysiology of MS comprises a number of processes including: (i) autoimmune inflammation, which likely starts in the periphery and subsequently also affects meninges and brain parenchyma, and is accompanied or followed by (ii) demyelination and various degrees of remyelination, axonal and neuronal damage, (iii) innate immune activation and inflammation, (iv) astrocyte activation and astrogliosis, and (v) oxidative stress and metabolic alterations in the CNS [1]. Currently available data indicates that autoreactive CD4+ T cells are activated in the periphery, which enables them to cross the blood-brain barrier, enter into the CNS and orchestrate downstream immune responses [1]. The specificity of these autoreactive T cells remains to be studied in further detail, but likely involves several possible autoantigens [6,7,8] as well as cross-reactivity to antigens from latent viruses [9] or the gut microbiome [10]. However, depending on the cytokine environment, CD4+ T cells develop into T helper (Th) cell subsets that are essential for host defense mechanisms [11]. In MS, IFNγ-secreting Th1 cells, IFNγ- and IL-17-secreting Th1*, and IL-17-secreting Th17 cells have been described to be pathogenic [12,13,14,15,16,17,18]. The high efficacy of B-cell depleting therapies indicates that proinflammatory B cells also play a key role in disease pathogenesis [19], and recent data have shown that autoreactive T cells and proinflammatory B cells activate each other and then proliferate spontaneously [6,7].

MS is diagnosed by clinical criteria in conjunction with magnetic resonance imaging findings and the demonstration of CSF-specific oligoclonal immunoglobulin G bands [20]. MS biomarkers that support the diagnosis and particularly those that better reflect the abovementioned disease mechanisms and allow monitoring of disease activity, treatment response, treatment-related adverse events or the prognosis of the future disease course are urgently needed [21,22,23,24].

During the last decades, the development of disease-modifying therapies for MS has been successful. All currently approved treatments modulate or suppress immune reactivity by depleting or interfering with the function and migration of adaptive immune cells, most importantly T and B cells. For patients presenting with highly active disease and refractory to the approved immunotherapies, autologous hematopoietic stem cell transplantation (aHSCT) recently became an option and is currently considered the most efficient treatment [25]. The rationale is to eradicate a dysfunctional immune system and then to induce an “immune reset” with a newly generated immune repertoire. Pathogenic autoreactive cells are eliminated along with all other immune cells by chemotherapy and anti-thymocyte globulin (ATG). Subsequently, the previously mobilized autologous hematopoietic stem and progenitor cells (HSCs) are re-infused and regenerate a new tolerant immune system [25]. This therapeutic concept has proven to be effective [25]. However, aHSCT is an invasive treatment with relevant treatment-associated complications. Since aHSCT targets the immune system, immunosuppression is an inevitable “on-target” effect. It involves numerous adverse effects secondary to the conditioning with chemotherapy and subsequent administration of ATG, particularly during the early phase after aHSCT. The most common ones are neutropenic fever, sepsis, gastroenteritis, urinary infection, reactivation of latent viral infections (cytomegalovirus (CMV), EBV, herpes simplex virus (HSV), varizella zoster virus (VZV)), and pneumonia [25,26,27]. Biomarkers for predicting these adverse events are currently missing [23].

Since aHSCT leads to a state free from further disease events in up to 90% of patients [28], one can assume that markers for the abovementioned pathomechanisms should normalize or improve. This rationale prompted us to examine candidate biomarkers for several processes relevant in MS with respect to their dynamics post-aHSCT.

Regarding processes leading to neuronal damage, several biomarkers have been studied in MS. Neurofilaments are exclusively present in the neuroaxonal cytoskeleton and have been used widely as biomarkers of ongoing axonal damage e.g., in neurodegenerative diseases or stroke. In these conditions, they are released into the extracellular space and subsequently into the CSF and, at lower concentrations, into the blood in case of neuronal damage or neuronal cell death [29]. Particularly neurofilament light chain (NfL) appears to accurately reflect acute axonal damage associated with inflammation but also chronic neuronal damage and has been developed into a validated biomarker in MS with high potential to become a clinically useful biomarker [22,29,30]. Several clinical aspects of MS correlate with CSF but also serum levels of NfL, e.g., disease activity, the degree of disability, the time since the last relapse in RRMS or response to treatment [29]. 

Glial fibrillary acidic protein (GFAP) is an astrocytic cytoskeletal intermediate filament protein released into the CSF and into the blood in disorders associated with astrocyte activation and astrogliosis following inflammation and neurodegeneration and therefore is highly expressed in MS lesions [31,32]. It is considered a validated CSF biomarker [22] for disease activity and progression in MS patients with predictive value for disability evolution [31,32,33,34]. In addition, GFAP serum levels appear to correlate with MS disease severity [35,36].

Another biomarker for MS is chitinase 3-like 1 (CHI3L1, also known as YKL-40) [22], a glycoprotein that is secreted mainly by activated macrophages and in the CNS also by reactive astrocytes [37,38]. Increased serum levels are associated with a variety of conditions characterized by chronic inflammation [23,39]. Its role in the CNS is not fully understood, but the occurrence of CHI3L1 in CNS inflammatory lesions suggests that it is part of the astrocytic response to regulate inflammation [40,41]. Consequently, CHI3L1 is described as a tissue remodelling factor and marker of neuroinflammation and macrophage as well as glial activation [23,37] and is discussed to be a suitable biomarker for MS disease activity and prognosis in the CSF [22,23,39,40]. In patients with a clinically isolated syndrome (CIS), elevated CHI3L1 is considered a risk factor for the conversion to MS or has been associated with faster disability progression in several studies [34,41,42,43,44]. 

Chemokines and their receptors are necessary for the recruitment of immune cells to specific tissues, thereby regulating leukocyte migration during homeostatic and inflammatory conditions [45]. Among these, the C-X-C motif chemokine ligand (CXCL) 13, a chemoattractant for B cells and CD4+ Th cells expressing its cognate receptor C-X-C chemokine receptor (CXCR) 5, is a promising inflammatory biomarker in the CSF in the context of Th1-mediated inflammatory processes [22]. It is elevated in the CSF of MS patients with active disease, has potential as a predictive marker for disease course or conversion from CIS to MS and reflects treatment response [46,47,48,49,50]. Moreover, increased CXCL13 levels in the serum correlated with active disease [51]. 

Other chemokines are considered exploratory biomarkers in MS [22]: Results of several studies demonstrate that CXCL9 and CXCL10 (also called interferon-γ-inducible protein 10, IP-10) are secreted in various diseases of the CNS, especially in neuroinflammation. CXCL9 and CXCL10 do not appear in non-lymphoid tissues under physiological conditions, but are induced by cytokines, particularly interferon-γ. As they bind to CXCR3, they mediate leukocyte trafficking of activated CD4+ Th1 cells, CD8+ T cells and NK cells following the CXCL9/10 gradient. In MS, CXCL9 and CXCL10 seem to mediate autoreactive T cells to CNS inflammatory lesions [6,52]. CXCL10 expression was detected in astrocytes and glial cells, amongst others, and seems to attract activated Th1 cells to the CNS as Th1 cells express CXCR3 [53,54]. There is some evidence for increased CXCL10 levels in serum and CSF in MS [55,56] but information on serum levels is limited.

In summary, we chose NfL as a biomarker for neuronal damage (ii), CHI3L1 as a marker of macrophage and glial activation and tissue remodelling (iii), and GFAP as a marker for astrocyte activation (iv). Additionally, we examined the chemokines CXCL9, 10 and 13 as markers of inflammation (i). Amongst them, CXCL9 and 10 are particularly associated with T cell recruitment of the Th1 phenotype and CXCL13 is involved in B cell recruitment, the two lymphocyte subsets we believe most relevant in MS pathophysiology. However, none of the abovementioned markers is disease-specific for MS and also not exquisitely cell- or process-specific and hence allows only limited conclusions as to the underlying disease mechanisms. 

## 2. Results

We studied the levels of several biomarkers in CSF of 11 MS patients and in serum of 32 MS patients who underwent aHSCT with the BEAM-ATG conditioning regimen as described previously [57]. For patient-related information, see Table 1. In general, patients showed an excellent clinical response during the time of follow-up as none of the patients showed new relapses or new MRI lesions. Concerning their EDSS development at the latest follow-up, few deteriorated (18.8%), while the majority improved in EDSS (59.3%) or stayed stable (21.9%).

In the CSF, median NfL levels increased non-significantly from 422 to 463 pg/mL 24 months following transplantation (Figure 1). However, the two patients with the highest NfL values in the CSF both decreased strongly 24 months following transplantation (patients 17 and 21, from 7504 to 339 pg/mL and from 1846 to 526 pg/mL), and also both improved in the EDSS after aHSCT until their latest follow-up. Solely GFAP showed an overall significant change in CSF, notably an increase 24 months after aHSCT. Other CSF markers (CHl3L1, CXCL10, CXCL9 and CXCL13) showed no significant changes. There was an insignificant decrease of CHI3L1 in the CSF 24 months after aHSCT. We measured a decrease in three patients while two patients showed an increase of CHI3L1 level in CSF after the transplantation. Amongst the non-significant changes of the other cytokines, CXCL9 and CXCL10 showed a slight increase 24 months after aHSCT. We did not detect CXCL13 in CSF from any of the eight patients tested, neither before nor after aHSCT. This was likely due to insufficient sensitivity of the ELISA (Figure 1).

In serum, we found that the neurodegeneration marker NfL and the astrocyte activation marker GFAP both significantly increased early after aHSCT and then normalised until the end of the observation period (Figure 2). NfL percentiles and z-scores showed the same dynamics as absolute NfL values (Appendix A). The two patients with a clear decrease of CSF NfL showed the same trend in the serum comparing pre- to 24 months post-aHSCT (from 78.7 to 9.3 and from 18.5 to 7.9 pg/mL). In the five patients, in whom we studied CHI3L1, we did not observe any significant changes. Serum CXCL10 levels were significantly increased one month after aHSCT and then regressed three months after aHSCT to stable levels at month 12 and 24 with insignificant changes. Non-significant effects on serum CXCL9 and CXCL13 levels were observed during follow-up. However, serum CXCL9 showed a trend for increase one month after aHSCT followed by stabilization at later time points. For serum CXCL13, a trend for an increase with a maximum at month 3 after transplantation was observed (Figure 2).

As CXCL10 serum levels were strongly increased early after aHSCT, a period where we did not observe any relapse activity, we hypothesized it might reflect immunological events early post-aHSCT. CMV as well as EBV reactivations occasionally occur in the context of the immunosuppression during aHSCT as described above. Further, all patients had various degrees of gastrointestinal symptoms. Interestingly, serum CXCL10 levels at month 1 after aHSCT were particularly increased in patients with a CMV viremia, while patients with an EBV viremia had normal CXCL10 levels (Figure 3A). However, patients without either CMV or EBV viremia also showed increases of CXCL10 from a median of 96 pg/mL pre-aHSCT to a median of 216 pg/mL one month post-aHSCT (Figure 2 and Figure 3A). Patients with CMV viremia (n = 3) had slightly increased serum NfL levels three months after aHSCT compared to patients without CMV viremia (n = 14) (Figure 3B).

CXCL10 is known to prime T cells towards a Th1 phenotype [58]. Using flow cytometry (see Figure 4A), we saw that EM CD4+ T cells were skewed towards a Th1 phenotype and subsequently less towards a Th2 phenotype. Interestingly, we found this to be the case not only after, but also prior to aHSCT (Figure 4B). Hence, we considered that the treatment(s) before aHSCT might play a role. Based on our data, CMV reactivation may be related to the pre-treatment as 7/8 patients or 87.5% with a CMV reactivation had an anti-CD20 pre-treatment. This reflects our general cohort in which most patients are pre-treated with anti-CD20 treatment (26/32 or 81.3%). We can, however, not rule out that CMV reactivation would not also occur with other pre-treatments. We also checked, if the Th subsets were different in patients with or without anti-CD20 treatment but found no significant differences. 

## 3. Discussion

In this study, we analysed the dynamics of MS relevant biomarkers pre- and post-aHSCT. Our cohort consisted of 32 individuals that underwent the procedure; however, some of the readouts were only performed in five patients representing a shortcoming of the reported results. Also important to mention, 26 of our 32 patients received anti-CD20 treatment before their aHSCT, four patients received natalizumab and two fingolimod (Table 1). In our studied cohort, we detected CMV viremia in eight and EBV viremia in nine out of 32 patients (four patients with CMV reactivation, five patients with EBV reactivation, and four patients with both CMV and EBV reactivation). This contrasts with one study using a milder regimen in a cohort not as substantially pre-treated with anti-CD20 medication, where EBV viremia occurred in 100% of the patients [59]. 

The quantification of CSF analytes is valuable due to its proximity to the CNS. Surprisingly, we observed a discrepancy in our biomarkers (Figure 1): while CHI3L1 remained at stable level, we detected a highly significant increase of GFAP two years after aHSCT. Both, CHI3L1 and GFAP, are described as markers of astrocyte activation or damage in MS as well as other neurological diseases [37]. However, whereas GFAP is referred to as a marker solely for astrocyte activation, CHI3L1 is described to indicate primarily innate inflammatory processes [60]. Hence, our results suggest that inflammation remains unchanged, while astrocyte activation is increased two years after aHSCT. Importantly, NfL did not increase, particularly not in patients with a high level of NfL pre-aHSCT, implying that the potential astrocyte activation is not correlated with an increased axonal or neuronal damage. This is in line with the literature, showing that pathological CSF NfL values tend to normalise after aHSCT [61]. Our markers of neuroinflammation, CXCL9 and 10 did not show marked changes, and CXCL13 could not be detected with the ELISA we had used. To conclude, we observe signs of sustained astrocyte activation based on increased GFAP in the CSF while neuroinflammation appears controlled after aHSCT, which is in line with the clinical outcome and absence of clinical or imaging signs of new neuroinflammation. 

Given that lumbar puncture is an invasive procedure, CSF was sampled only twice, whereas serum could be collected at several time points already early after aHSCT, allowing for a better understanding of the dynamics. The CNS derived biomarkers, NfL and GFAP, showed a strong increase and then a trend back to pre-aHSCT values in the serum (Figure 2), indicating that the transplantation may lead to acute neuronal injury and reactive astrocyte activation that resolved with time. The marked increase of NfL and GFAP one month after aHSCT might also have been seen in the CSF but we could not access it due to sample restrictions. In previous studies, serum NfL had not been measured as early as one month after aHSCT. It was reported to decrease later after aHSCT compared to pre-aHSCT [62,63], a finding we could not reproduce. However, these groups used different treatment regimens. More in line with our results was a measure of neurofilament heavy chain (NfH) after autologous bone marrow transplantation (BMT), where a significant increase in NfH in the serum was seen one until 12 months post-BMT [64]. Regarding serum GFAP, one group had previously reported an increase three months after aHSCT [63]; therefore, they presumably missed the early marked increase one month post-aHSCT. As none of the patients had signs of neuroinflammation, this appears an unlikely reason for the neuronal damage. The most likely cause for the transient NfL increase is damage to neurons or axons by the chemotherapeutic agents. In addition, the finding of an increased MRI-measured atrophy rate of the brain early post-aHSCT [64,65,66,67] indicates acute neurotoxic effects of aHSCT. This is not surprising as two out of the four cytotoxic drugs used in the BEAM-ATG protocol efficiently penetrate the intact blood-brain barrier and act on brain cells [68]. Another possible reason could be an alteration in the gut microbiota due to chemotherapy and the antibiotic prophylaxis and treatments, leading to an imbalance in the short chain fatty acid profile. 

Of further interest, we observed a clear increase of CXCL10 in the serum early after aHSCT (Figure 2). This is in line with findings of one other report [69], although a slightly less intensive treatment protocol had been used in that study. CXCL10 and its receptor CXCR3, which is expressed by potentially encephalitogenic T cells, are highly relevant in the pathogenesis of MS [55]. Beyond that, CXCL10 is known to be increased in patients suffering from CMV infection or reactivation, e.g., after lung transplantation ([70] and reviewed in [71]), in elderly [72], in corneal CMV infection [73], septic patients [74], as well as in pregnancy [75]. Solely after kidney transplantation, CMV reactivation was reported without an increase in CXCL10 [76,77]. Hence, our finding that CMV-reactivating patients show strongly increased serum CXCL10 levels (Figure 3A) likely stems from the CMV reactivation and not from increased MS-related inflammation. CXCL10 may, however, also be released in the context of other treatment-related infections, particularly those of the gut. This could explain the remaining difference in serum CXCL10 pre- to early post-aHSCT present in patients without viral reactivation. In contrast to CMV, EBV does not appear to lead to increased serum CXCL10, as patients suffering from EBV reactivation did show similar CXCL10 values as patients without. Several patients suffered from other treatment-related infections, particularly gastroenteritis, and therefore it is difficult to assign elevated CXCL10 levels unambiguously to a single infection. 

Since CXCL10 plays a prominent role in Th1-mediated immune responses, we were interested if the parallel increase of NfL and CXCL10 in patients with CMV viremia was associated with changes of T helper phenotypes of CD4+ EM T cells. Indeed, there was an increase in Th1 and decrease in Th2 in patients with a CMV viremia (Figure 4). For CD8+ T cells, increased CMV-specific cells were found on several occasions with incomplete CMV control. For example, CMV-specific CD8+ T cells were reported to be increased in patients with a CMV reactivation after allogeneic stem cell transplantation [78] and in chronic lymphocytic leukemia patients having a high risk of CMV reactivation [79]. However, this needs further study, particularly regarding CMV-specific CD4+ T cells [80].

Moreover, we considered that particularly CNS-resident microglia were reported to produce CXCL10 and lead to lymphocyte migration upon CMV stimulation [81]. This, together with our findings of increased serum CXCL10 and NfL as well as Th1 EM CD4+ T cell phenotype in patients with CMV viremia (Figure 3A,B and Figure 4B), might indicate that these are related. In fact, CXCL10 is secreted by endothelial cells upon CMV infection and recruits particularly effector T cells [82]. Interestingly, CMV seropositivity was discussed as a protective factor against the development of MS [83], although the mechanisms, by which this could be mediated, are not understood. Hence, it remains unclear if CMV is beneficial or exerts negative effects in MS patients after aHSCT. 

Overall, our results cannot explain, which mechanism leads to the excellent results of aHSCT in MS, as there was no drastic decrease of MS-related biomarkers after aHSCT. Except for NfL, the lack of a clear cell specificity is another factor that renders it difficult to extrapolate from the biomarkers that we examined in this study to specific cellular and molecular mechanisms. With relation to CNS integrity, we conclude that aHSCT transiently leads to neuronal damage as measured by increased NfL, which is resolved after several months. Astrocyte activation in the CNS appears increased for longer periods, though it was not associated with tissue damage after the first months. It is therefore difficult to interpret, which mechanisms are related to the increased GFAP. Inflammatory markers such as serum CXCL10 were strongly increased shortly after aHSCT. This is likely associated with CMV reactivation or other treatment-related infections, most likely of the intestinal system, but not to flares in MS activity. 

## 4. Material and Methods

### 4.1. Patients, Treatment Regimen and Samples

MS patients suitable for the aHSCT were included in the aHSCT-in-MS registry. Inclusion criteria enclosed age between 18 and 55 years, EDSS between 2 and 6.5, as well as breakthrough disease activity defined as (1) clinical activity (clinically confirmed MS relapse), (2) MRI activity (contrast-enhancing or new T2-hyperintense lesions on brain or spinal cord MRI), and/or (3) confirmed disability progression based on increases in EDSS, sustained for at least 6 months, despite treatment with a least one highly effective disease-modifying therapy. All patients provided written informed consent. Ethic protocols for the collection of biomaterials (blood and CSF) and for studying aHSCT are found under BASEC-No. 2018-01854. 

The mobilization regimen comprised cyclophosphamide 2 × 2 g/m^2^ and G-CSF, and CD34+ HSCs were collected thereafter by leukapheresis. The conditioning regimen followed the BEAM-ATG protocol as published previously [57], containing BCNU (carmustine), etoposide, ara-C (cytarabine), and melphalan. The re-transfusion included CD34+ autologous HSCs (between 3−8 × 10^6^ CD34+ cells/kg body weight). As an in vivo depletion of possibly transfused T cells, rabbit ATG was administered after CD34+ cell re-transfusion. Patients were defined to have a CMV or EBV reactivation when the respective virus could be detected in the peripheral blood by PCR, i.e., were viremic. 

Moreover, peripheral blood mononuclear cells (PBMCs) were isolated freshly from anticoagulated blood using Ficoll-Paque density gradient. PBMCs were then cryopreserved for 24−48 h at −80 °C followed by long-term storage in liquid nitrogen. 

### 4.2. Detection of Biomarkers

All biomarkers were measured by enzyme-linked immunosorbent assays with commercial ELISA kits or by single-molecule array assays (SIMOA). Within assay (intra-assay) coefficients of variation (CV) of each measurement are shown in Appendix A.

NfL was measured using the Simoa^®^ Human Neurology 2-Plex B assay Nf-L, GFAP (Quanterix, Billerica, MA, USA) on the HDx system following the instructions. Samples were diluted 4× and the limit of detection was 0.065 pg/mL (range 0.040–0.098 pg/mL). None of the samples were below this threshold. Additionally, we calculated NfL percentiles and z-scores [30]. The intra-assay CV was 3.7%. Between assays (inter-assay) precision was assessed on three human serum quality controls (one spiked with human cerebrospinal fluid) with CVs rating from 4.7 to 8.9%.

GFAP was quantified by the same SIMOA technique, i.e., Simoa^®^ Human Neurology 2-Plex B assay Nf-L, GFAP (Quanterix, Billerica, MA, USA) on the HDx system, after a dilution of 4×. The detection limit was 0.475 pg/mL (range 0.135–0.26 pg/mL) and none of the samples were below it. The intra-assay CV was 4.9% and inter-assay CVs from 0.9 to 6.5%.

For detection of CHI3L1 (YKL-40) we used the MicroVue YKL-40 EIA kit according to manufacturer’s protocol (Quidel, San Diego, CA, USA). All samples were measured in duplicates. The limits of quantification were between 15.6 and 300 ng/mL and the limit of detection was 5.4 ng/mL. The supplier reports an intra-assay CV from 5.6 to 6.6% and inter-assay from 6.0 to 7.0%. 

CXCL9, CXCL10 and CXCL13 were determined in serum and in CSF by Human Quantikine ELISA kits following the manufacturer’s directions (R&D Systems, Minneapolis, MN, USA) for serum. For CXCL9, we used the Quantikine ELISA Human CXCL9 with a detection limit of 3.84 pg/mL and measuring range of 31.3–2000 pg/mL. The CV is indicated from 3.1 to 3.9% (intra-assay) and from 5.2 to 8.2% (inter-assay). The Quantikine ELISA for Human CXCL10 had a detection limit of 1.67 pg/mL and a measuring range of 7.8–500 pg/mL. The CV intra-assay is reported from 3.0 to 4.6% and inter-assay from 5.2 to 8.8%. For CXCL13 (Quantikine ELISA Human CXCL13) the minimum detectable dose was 1.64 pg/mL and the measuring range 7.8–500 pg/mL. The supplier stated an intra-assay precision from 2.7 to 4.4% and inter-assay from 8.7 to 9.6%. All samples were measured undiluted and in duplicates in the initial run. In case of CXCL10, five samples showed an overflow and hence were diluted with a dilution factor of three and then re-run.

### 4.3. Flow Cytometry Staining

In brief, PBMCs were thawed, counted and 1 million cells analyzed. First, they were stained in PBS with a viability marker fixable green (Invitrogen Thermo Fisher Scientific, Waltham, MA, USA) while blocking with 40 μg/mL of human purified IgG (Merck, Sigma Aldrich, Schaffhausen, CH). Next, PBMCs were stained in FACS buffer (PBS containing 1% FCS and 2 mM EDTA) with the following surface markers: CD3 in AF700 (BD), CD4 in PE-Texas Red (Invitrogen), CD8 in BV510, CD45RA in BV711, CCR7 in BV421, CCR6 in BV785, CCR4 in APC, CRTh2 in PE (all BioLegend, San Diego, CA, USA). Samples were acquired with a BD LSR Fortessa (BD, Franklin Lakes, NJ, USA) and data analyzed using the software FlowJo (FlowJo Ashland, OR, USA). 

### 4.4. Statistics

To compare two groups, we used Wilcoxon test. In case of more than two groups (serum), we used first a global Kruskal-Wallis test and then a paired Wilcoxon signed-rank test as a post-hoc test with the pre-aHSCT time point as reference group. Significance levels were * for *p* ≤ 0.05, ** for *p* ≤ 0.01, *** for *p* ≤ 0.001 and **** for *p* ≤ 0.0001. Whenever boxplots are displayed, the thick line in the middle represents the median, the lower and upper hinges show the first and third quartiles (the 25th and 75th percentiles), the whiskers range from the hinge to the smallest or largest value no further than 1.5 times the distance between the first and third quartiles. Statistical analysis and visualizations were all performed in R Core Team (2022).

## Figures and Tables

**Figure 1 ijms-23-10946-f001:**
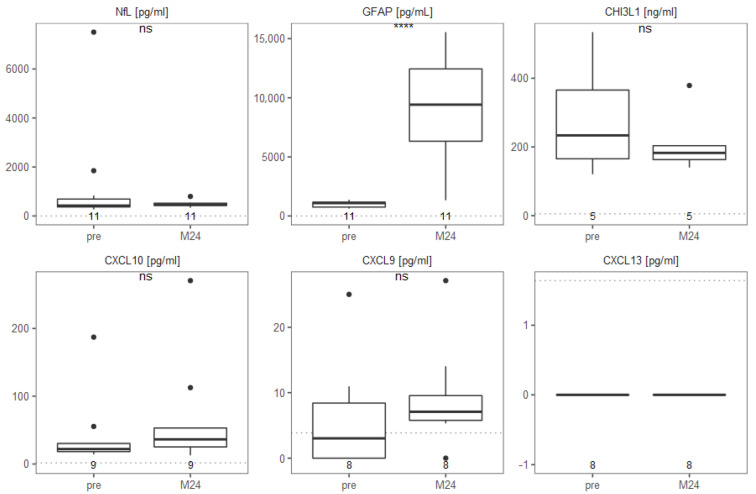
GFAP is increased in CSF after aHSCT while other MS biomarkers appear unchanged. Biomarker levels in the CSF pre- and 24 months post-aHSCT in MS patients. Concentrations in pg/mL of NfL, GFAP, CXCL9, CXCL10 and CXCL13 and in ng/ml of CHI3L1 are shown. Detection limits are visualized with a dotted line, number of patients are indicated below each boxplot. Wilcoxon test was used to detect significance, **** for *p* ≤ 0.0001.

**Figure 2 ijms-23-10946-f002:**
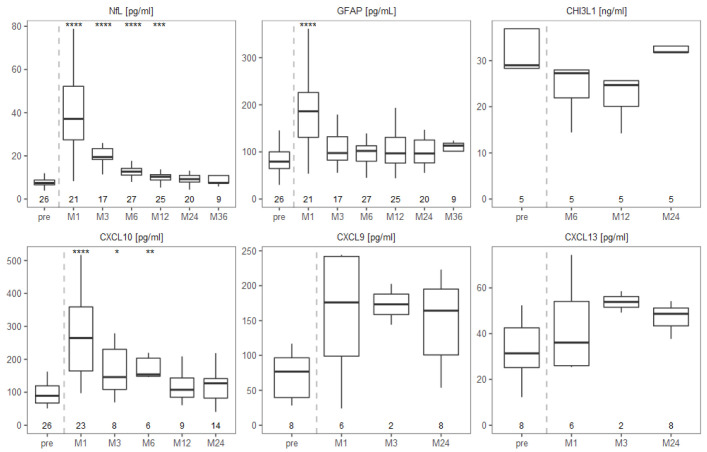
Serum levels of NfL, GFAP and CXCL10 transiently increase after aHSCT and then trend back to pre-aHSCT levels. Levels of studied biomarkers in the serum before and at several time points after aHSCT in MS patients. Concentrations in pg/mL are shown for NfL, GFAP, CXCL10, CXCL9 and CXCL13. Concentrations for CHI3L1 are shown in ng/ml. Numbers at the bottom indicate number of samples included in respective boxplot. Global Kruskal-Wallis test was used first and in case of a significant dif-ference, then a paired Wilcoxon signed-rank test as a post-hoc test with the pre-aHSCT time point as reference group. Significance levels were * for *p* ≤ 0.05, ** for *p* ≤ 0.01, *** for *p* ≤ 0.001 and **** for *p* ≤ 0.0001.

**Figure 3 ijms-23-10946-f003:**
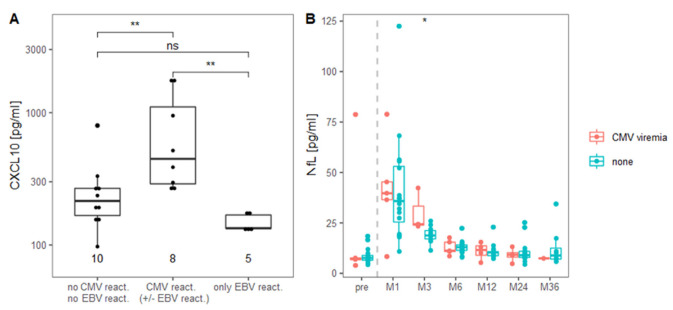
Patients suffering from a CMV viremia post-aHSCT present with higher serum CXCL10 one month post-aHSCT and higher serum NfL three months post-aHSCT. (**A**) CXCL10 in pg/mL in the serum one month after aHSCT in MS patients distinguished by viral reactivation of CMV (of CMV exclusively (n = 4) or simultaneously of CMV and EBV (n = 4)) vs. viral reactivation of only EBV (n = 5) vs. no CMV nor EBV reactivation (n = 10). (**B**) NfL in pg/mL in the serum at several time points after aHSCT differentiating between patients with and without CMV reactivation. In (**A**), a global Kruskal-Wallis test was used fist. To find differences between groups, an unpaired Wilcoxon signed-rank test was performed with * for *p* ≤ 0.05, and ** for *p* ≤ 0.01.

**Figure 4 ijms-23-10946-f004:**
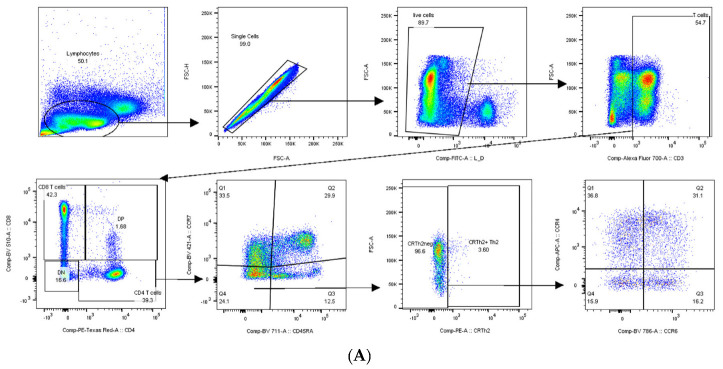
Effector-memory (EM) T cells show a higher Th1 and lower Th2 phenotype in patients with CMV viremia after aHSCT. PBMCs were stained with fluorophore-conjugated antibodies and analyzed by flow cytometry. In (**A**) the gating strategy is depicted in one representative example. (**B**) Th1 and Th2 phenotypes of EM CD4+ T cells before and at 1, 3, 6, and 12 months after aHSCT. For CMV viremia group, n = 5, 5, 4, 5, 4, for no CMV viremia group, n = 21, 16, 15, 22, 21. To detect significant differences, an unpaired Wilcoxon signed-rank test was used, significance levels were * for *p* ≤ 0.05, and ** for *p* ≤ 0.01.

**Table 1 ijms-23-10946-t001:** Patient demographics. Patient characteristics are depicted including CMV and EBV reactivation status. Patients 7, 17, 18 and 27 showed signs of gastroenteritis, which required transient hospitalization in patients 7, 17, and 18, and all four transiently experienced an Uhthoff phenomenon.

	Age	Sex	Diagnosis	Last Therapy before aHSCT	EDSS at aHSCT	CMV Viremia	EBV Viremia
aHSCT_MS_01	38	female	RRMS	Natalizumab	4		
aHSCT_MS_02	36	male	RRMS	Fingolimod	5.5		
aHSCT_MS_03	53	female	PPMS	Fingolimod	6.5		
aHSCT_MS_04	32	female	RRMS	Rituximab	4		
aHSCT_MS_05	47	male	PPMS	Rituximab	6.5		
aHSCT_MS_06	49	female	PPMS	Rituximab	4		
aHSCT_MS_08	30	male	PPMS	Ocrelizumab	4.5		
aHSCT_MS_09	48	female	SPMS	Rituximab	6		
aHSCT_MS_10	36	female	SPMS	Ocrelizumab	4.5		
aHSCT_MS_11	47	male	RRMS	Ocrelizumab	3		
aHSCT_MS_12	45	male	SPMS	Rituximab	6.5		
aHSCT_MS_13	41	male	PPMS	Ocrelizumab	3		
aHSCT_MS_15	54	female	SPMS	Rituximab	5.5		
aHSCT_MS_16	33	female	RRMS	Ocrelizumab	4.5		
aHSCT_MS_19	43	male	PPMS	Rituximab	6		
aHSCT_MS_20	47	male	PPMS	Ocrelizumab	6		
aHSCT_MS_22	54	female	SPMS	Rituximab	4.5		
aHSCT_MS_24	44	female	SPMS	Rituximab	4		
aHSCT_MS_25	44	male	SPMS	Ocrelizumab	3.5		
aHSCT_MS_21	25	male	RRMS	Ocrelizumab	2		x
aHSCT_MS_23	43	male	RRMS	Natalizumab	3		x
aHSCT_MS_26	40	female	RRMS	Natalizumab	3.5		x
aHSCT_MS_28	32	female	RRMS	Ocrelizumab	2		x
aHSCT_MS_36	39	female	RRMS	Ocrelizumab	3		x
aHSCT_MS_07	33	female	SPMS	Rituximab	5	x	
aHSCT_MS_14	25	male	PPMS	Rituximab	6.5	x	
aHSCT_MS_27	51	male	SPMS	Ocrelizumab	4.5	x	
aHSCT_MS_31	40	female	RRMS	Ocrelizumab	3.5	x	
aHSCT_MS_17	39	female	RRMS	Ocrelizumab	2.5	x	x
aHSCT_MS_18	47	male	RRMS	Natalizumab	4	x	x
aHSCT_MS_32	35	male	RRMS	Ocrelizumab	3	x	x
aHSCT_MS_34	29	female	RRMS	Ocrelizumab	2	x	x

## Data Availability

Not applicable.

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
