# Peer review of "Dynamics of Inflammatory and Neurodegenerative Biomarkers after Autologous Hematopoietic Stem Cell Transplantation in Multiple Sclerosis"

_ijms, 2022, doi:10.3390/ijms231810946_

Round 1

Reviewer 1 Report

This original research article indirectly investigates the mechanism of action of aHSCT by evaluating serum and CSF biomarkers relevant to the pathophysiology of MS. The article is generally well written and presents useful data for the field of aHSCT research. However, the authors present biomarker data with very minimal reference to clinical outcomes and some arguments made by the authors appear contradictory (please see below). It is challenging to interpret these indirect markers for MS activity even with reference to clinical and MRI findings, let alone without. The authors seem to have access to this data and it would be sensible to make the most out of this. These clinical data would greatly improve the quality and relevance of manuscript.

Major concerns/improving clarity of the manuscript.

Line 219: it would be helpful to add that this was 24 months following aHSCT. Line 220: Was this also 24 months for the two patients with markedly reduced NfL CSF values and did these patients improve clinically/on MRI? Was this also reflected in the serum? It is strange that two patients had this huge decrease in CSF NfL but then the next paragraph states that in general NfL serum levels increased. Perhaps the long time period between CSF sampling compared to the short intervals between serum sampling could go some way towards explaining this, as NfL levels eventually normalised in the serum. Could the authors acknowledge and explain this in the manuscript? Perhaps in these two patients serum NfL was decreased.

Line 253. The Th1 skewing that was also present before aHSCT may be due to, e.g., B cell therapies in these patients. Was the previous medication considered in this analysis? Did a specific therapy have an associated effect on T cell phenotype and the occurrence of CMV viremia. 

The authors state “To conclude, we observe signs of sustained astrocyte activation based on increased GFAP in the CSF while neuroinflammation appears controlled after aHSCT.” without including MRI or clinical data to support that the patients did not experience CNS inflammation or relapse after aHSCT. Without this data and based only on indirect biomarker evaluation, I do not believe one can say that neuroinflammation appears controlled. Or does the clinical data support this?

Line 326. The authors state “One speculation is that the massive activation by CMV deviates the immune system away from autoreactivity in the CNS, as none of the studied patients developed any new inflammatory lesions.” Do they mean that none of the patients with increased serum CXCL10 and NfL with a Th1 EM CD4+ T cell phenotype with CMV viremia or do they mean all 35 patients included in the study? In addition, the increased NfL levels would indicate axonal damage, which does not really quite fit with the patients having no inflammatory activity. It is also not clear whether CMV positivity is a protective factor against MS development. CMV positivity has been associated with a decreased MS risk in the referenced study but this is yet to be fully understood and the authors of this referenced study actually state that it remains to be shown whether this is truly a protective effect. What indicates that these patients had “massive” CMV activation? High viral titres? Severe symptoms? The immune response? In addition, this statement is contradicted by the fact that on lines 248 to 249 the authors state that the three patients that had CMV viremia and had increased NfL, suggesting neuronal damage. Why in the discussion is it then stated that 8 patients had CMV viremia (lines 261 to 262)? In addition, it is not clear how the authors differentiated between CMV viremia and CMV reactivation (the same can be said for EBV). It is also unclear from the graph whether these patients with CMV reactivation also had EBV activation, as Figure 3A has “CMV react. +/- EBV react.” but no column for “only CMV react.” Was there a technical reason for this? Finally, what do the authors mean by “deviates the immune system away from autoreactivity”? Could they suggest a mechanism to support this claim? The authors then suggest that screening for CMV seropositivity appears indicated, but do not suggest why. Are they suggesting CMV reactivation has prognostic value, as the previous text might suggest? Or are they suggesting that the risk for CMV reactivation as subsequent illness is important to establish – if so, should we also be screening for EBV?

Minor concerns/English recommendations.

1.       Line 10. …is a highly efficient therapy of multiple sclerosis à …is a highly efficient treatment for multiple sclerosis

2.       Line 51. …pathogenic. Besides, particularly the high à (delete: Besides, particularly) …pathogenic. The high

3.       Line 233. ..in the five patients, in which we studied CHI3L1 à n the five patients, in whom we studied CHI3L1

Reviewer 2 Report

This article investigates the dynamics of inflammatory and neurodegenerative biomarkers after autologous hematopoietic stem cell transplantation in multiple sclerosis. Although the topic is interesting, in my opinion the context of the research is not sufficiently described, the results are not adequately presented and the discussion in several parts is a mere repetition of the results.

Specific comments:

- The Abstract should be focused on the main findings of the research. Moreover, non-standard or uncommon abbreviations/acronyms (if necessary) must be defined at their first mention in the abstract itself (for example ELISA, SIMOA, GFAP, CNS...). A concluding sentence reflecting the scientific contribution of the study should be added at the end of the abstract.

- The "Materials and Methods" section: Intra-assay and inter-assay coefficients of variation values of all measurements should be added.

- Statistics: It should be explained whether a normality test is applied or not. The normality test should be added to statistical analysis.

- In my opinion the "Discussion" section in the present form is a repetition of the results. If possible, this section should critically analyse and interpret the findings of the study. All results should be connected to each other, the mechanism of action of autologous hematopoietic stem cell transplantation and the relation structure-activity in multiple sclerosis should be analysed and discussed by comparing the observed effects with those previously observed for other related theraphy in similar models.

Round 2

Reviewer 2 Report

Dear authors,

I understand that you have made the changes I suggested in the article. Good luck